# Theoretical study of ice cover phenology at large freshwater lakes based on SMOS MIRAS data

Vasiliy Tikhonov [1,2], Ilya Khvostov [3], Andrey Romanov [3], Evgeniy Sharkov [1]

[1]Department of Earth Research from Space, Space Research Institute of the Russian Academy of Sciences, 84/32 Profsoyuznaya Str., Moscow 117997, Russia
[2]Moscow Institute of Physics and Technology (State University), 9 Institutskiy Per., Dolgoprudny, Moscow Region 141701, Russia
[3]Institute for Water and Environmental Problems, Siberian Branch of the Russian Academy of Sciences, 1 Molodezhnaya Str., Barnaul 656038, Altai Krai, Russia

*Correspondence to*: Vasiliy Tikhonov (vtikhonov@asp.iki.rssi.ru)

**Abstract.** The paper presents a theoretical analysis of seasonal brightness temperature variations at a number of large freshwater lakes - Baikal, Ladoga, Great Bear Lake (GBL), Great Slave Lake (GSL), and Huron - retrieved from Microwave Imaging Radiometer with Aperture Synthesis (MIRAS) data (1.4 GHz) of the Soil Moisture and Ocean Salinity (SMOS) satellite. The analysis was performed using the model of microwave radiation of plane layered heterogeneous nonisothermal medium. The input parameters for the model were real regional climatological characteristics and glaciological parameters of ice cover of the study lakes. Three distinct seasonal brightness temperature time regions corresponding to different phenological phases of the lake surfaces: complete ice cover, ice melt and deterioration, and open water, were revealed. The paper demonstrates the possibility to determine the beginning of ice cover deterioration from satellite microwave radiometry data. The obtained results can be useful for setting the operating terms of winter crossings and roads on ice, since with the beginning of ice deterioration, these transportation routes across water bodies (rivers, lakes, water reservoirs) become insecure and cannot be used any more.

## 1. Introduction

Satellite microwave radiometry is widely employed in studies of the Earth's cryosphere (Tedesco, 2015). The data are used to retrieve sea ice concentration and thickness in polar regions (Tian-Kunze et al., 2014; Tikhonov et al., 2016; Ricker et al., 2017), investigate glacial depths of Greenland and Antarctica (Zwally and Giovinetto, 1995; Magand et al., 2008), analyze ice cover characteristics (Dietz

et al., 2012; Lemmetyinen, 2012), detect permafrost zones and estimate depth of soil freezing (Zuerndorfer et al., 1990). They are also used to assess the depth and phenology of ice cover of large freshwater lakes (Kang et al., 2010, 2012, 2014; Kang, 2012; Kontu et al., 2014; Du et al., 2017).

Kang et al. (2010, 2012, 2014) analyzed seasonal variations of brightness temperature at GBL and GSL derived from AMSR-E (Advanced Microwave Scanning Radiometer for EOS) data. They demonstrated that the 18.7 GHz H-pol channel was the most sensitive to the phenology of ice cover. The channel data allow determining the dates of ice cover freeze-onset/melt-onset and ice-on/ice-off (Kang et al., 2010; Kang, 2012; Kang et al., 2012). It was found in (Kang et al., 2010; Kang, 2012; Kang et al., 2014) that at 18.7 GHz V-pol, brightness temperature greatly depends on ice thickness. The results obtained in these works enabled the authors to develop an algorithm for the determination of phenology phases of lake ice cover from 18.7 GHz H-pol (Kang, 2012; Kang et al., 2012) data, as well as calculate simple regression dependencies of ice thickness on 18.7 GHz V-pol brightness temperature (Kang, 2012; Kang et al., 2014) for GBL and GSL.

Du et al. (2017) presented a novel automatic algorithm for daily analysis of ice condition of the lakes of the Northern Hemisphere. The data used are AMSR-E and AMSR2, channel 36.5 GHz, H-pol.

However, neither AMSR-E, nor the newer AMSR2 encompass all the possibilities of satellite microwave radiometry. Their lowest frequency channel is 6.9 GHz H- and V-pol (Tikhonov et al., 2016). Even at this frequency, the penetration depth in wet snow and ice is only a few centimeters or less (Tikhonov et al., 2013, 2014). So, during the melt period, AMSR receives radiation formed only by the thin upper layer of the medium (snow or ice) and no signal from deeper layers (ice cover). Besides, the lower layer of ice cover in direct contact with water is always wet. This also entails the loss of information on the characteristics of deep ice layers and inadequate interpretation of satellite data even in winter (cold) period.

The MIRAS instrument on board the SMOS satellite operates at 1.4 GHz (Kerr et al., 2010). At this frequency, the electromagnetic penetration depth is considerably greater (Tikhonov et al., 2013, 2014) and, in contrast to AMSR, the radiation emitted from deeper layers of snow and ice cover can be received. In recent years, a considerable number of publications have appeared regarding studies of various Earth covers in L-band. Among them are works on: brightness temperature variations of the

Antarctica ice sheet (Macelloni et al., 2013, 2014); analysis of snow cover brightness temperature dependencies on snow wetness and density and ground permittivity (Naderpour et al., 2017; Schwank et al., 2015; Schwank and Naderpour, 2018a, 2018b); snow thickness retrieval over thick Arctic sea ice (Maaß et al., 2013); brightness temperature variations of Arctic sea ice (Heygster et al., 2009; Richter et al., 2018); retrieval of Arctic sea ice thickness (Huntemann et al., 2014; Kaleschke et al., 2012, 2016; Richter et al., 2017; Tian-Kunze et al., 2014; Zhou et al., 2018).

Various models of radiative and dielectric properties of layered media (snow cover, sea and lake ice, ice sheet, etc.) are employed to interpret data obtained by satellite microwave radiometers (see, for example, Heygster et al., 2009). The most widely used are HUT model (Helsinki University of Technology) (Pulliainen et al., 1999; Lemmetyinen et al., 2010) and MEMLS (Microwave Emission Model of Layered Snowpacks) (Matzler, Wiesmann, 1999; Wiesmann, Matzler, 1999). HUT model was successfully used to analyze brightness temperature of snow- and ice-covered lakes and wetlands derived from aerial and satellite microwave radiometry data (Gunn et al., 2011; Kontu et al., 2014). Modified versions of MEMLS were used for detailed analysis of L-band emission of freezing ground covered in snow as well as wet snow cover (Schwank et al., 2014; Schwank et al., 2015; Schwank and Naderpour, 2018a, 2018b).

The paper presents a theoretical analysis of seasonal variations of ice cover of a number of freshwater lakes based on MIRAS L-band data. We use the model of thermal emission of multi-layered non-isothermal media (Sharkov, 2003). The dielectric properties of each layer are calculated by a quasi-wave model of effective dielectric constant of a multi-phase dispersed medium (Boyarskii et al., 1994, 2002). Both models were found effective for the retrieval of brightness temperature of sea ice (Tikhonov et al., 2013, 2014), snow-firn layers of Antarctica (Tikhonov et al., 2017), as well as snow cover (Tikhonov et al., 2008). Here, the former model is modified for freshwater lakes. Using L-band allows receiving thermal emission from deeper ice layers in contrast to high frequency channels of SSM/I, SSMIS, AMSR-E and AMSR2. Its analysis provides information on structural and physical changes practically along the whole column of snow cover and lake ice.

## 2. Data description

Five large freshwater lakes located in different regions of the world were selected for the study. They are lakes Baikal and Ladoga situated in different climatic zones of Russia, GBL and GSL in the northeast of Canada, and Huron, one of the Great Lakes of North America (Table 1).

### 2.1. Satellite data

To determine surface brightness temperature at 5 K accuracy, the SMOS L1C v620 product (Gutierrez et al., 2014) was used. L1C data are georeferenced to the DGG ISEA 4H9 discrete hexagonal geodesic grid (Sahr et al., 2010). The grid cells chosen for each lake are shown in Fig. 1 with all center coordinates listed in Table 1. The cell linear size is 15 km. L1C is produced from SMOS MIRAS data of 21 cm wave length (frequency 1.4 GHz) and resolution ~35x50 km for H-pol and V-pol at an angle of 42.5°. The cells were chosen so that the antenna pattern did not reach the coast.

### 2.2. Climatic and glaciological data

Regional climatic data of the lakes and glacilogical characteristics of their ice covers were collected from a number of publications and websites. They provided multi-year air and water temperature, ice thickness and snow on ice thickness observational data for all the study lakes. As to Baikal and Ladoga, these were Russian language works of Galaziya (1984), Tulokhonov (2008), Rumyantsev et al. (2012), Rumyantsev (2015), as well as websites of the Institute of Lake Science of the Russian Academy of Sciences (http://www.limno.org.ru/win/rlake.php) and  the Climatic Reference Book of the Settlements of Russia (http://atlas-yakutia.ru/weather/spravochnik/temp/climate_sprav-temp_3043201321.php;    http://atlas-yakutia.ru/weather/spravochnik/sndp/climate_sprav-sndp_229120574.php). For GBL and GSL, we used data of Woo et al. (2007), Rouse et al. (2008), Kang et al. (2010, 2012, 2014), and Kang (2012), as well as websites of the Canadian Cryospheric Information Network (https://www.socc.ca/index.php/ccw/lakeice/links) and Polar Data Catalogue (https://www.polardata.ca/pdcsearch/), Governement of Canada: Historical Data and Canadian Climate Normals        (http://climate.weather.gc.ca/historical_data/search_historic_data_e.html;        and

 In case of Huron, data provided by Rumyantsev et al. (2012) and NOAA websites (https://www.glerl.noaa.gov//metdata/; https://coastwatch.glerl.noaa.gov/statistic/statistic.html) were used.

Because the lakes have considerable dimensions, we sought climatic and glaciological data
sources that were geographically close to the areas of the chosen SMOS L1C grid cells. For Baikal, it is the settlement of Davsha, for Ladoga, they are Valaam Island, the town of Olonets and Pitkyaranta settlement); for GBL, the Déline settlement; for GSL, the town of Yellowknife; for Huron, Bay Island and the town of Alpena. Multi-year ground data on air and water temperature, ice and snow cover thicknesses were averaged and approximated by smooth dependencies (Figs. 2-4).

Figure 2 presents average seasonal variations of air temperature for the five study regions. It is clear from the figure, that GBL is located in the coldest region and Huron in the warmest one.

Figure 3 shows averaged seasonal variations of water surface temperature in the central part of each lake. GBL is the coldest. Even in summer, its surface water temperature does not exceed $7^{°}$ C. GBL remains free of ice for a very short period that seldom begins before end of June.  The warmest
lake is Huron, its water surface temperature is positive nearly all the time. Huron almost never freezes up completely.

Figure 4 presents average ice thickness growth for four lakes that get completely covered by ice in the cold period of year. According to the plots, the most thick ice cover is formed on Baikal. Formation of stable ice cover begins in mid-December. Ice covers of GBL and GSL last for longer
periods, starting from end of October.  On Ladoga, ice thickness is much less compared to the others and the ice cover becomes stable as late as February. On all the lakes, the thickness of snow cover above ice does not exceed 50 cm. GBL has the thickest snow cover. A common feature for all lakes is that snow thickness decreases with the approach of warm season.

## 2.3. Satellite lake data of 2012-2013

For all lakes, 2012-2013 seasonal dependencies of brightness temperature in the selected grid cells were examined. From the analysis of these data for Baikal, Ladoga, GBL and GSL, three Time Regions (TRs: TR1, TR2, TR3) clearly distinct in brightness temperature characteristics were identified.

For Huron, no dramatic seasonal variations of brightness temperature were recorded, its evolution fully corresponded to TR1 observed at the other four lakes (Fig. 5).

Examination of climatic and glaciological conditions of the study regions (Figs. 2-4) showed that TR1 was associated with ice-free water surface, TR2 stable ice cover, and TR3 spring evolution of

5 lake ice.

## 3.  Model of microwave radiation from freshwater lake

### 3.1  The model

A theoretical calculation was performed to interpret and analyze brightness temperature variations in the selected regions of the lakes. The model of thermal emission of multi-layered non-

10 isothermal media (Sharkov, 2003; Tikhonov et al., 2014) modified for freshwater lakes was used. The model was developed on the basis of radiative transfer theory for a medium composed of flat layers taking into account multiple reflection of radiation at every layer boundary (Sharkov, 2003). Each layer was assumed to have specific physical and structural characteristics: composition, temperature, density, volumetric wetness, etc. According to the model, the radiative system consists of lake water surface

covered with lake ice and then snow with a thick layer of atmosphere above (Fig. 6). For such system, brightness temperature $T_{Br}$ is determined from the following relations (Sharkov, 2003; Tikhonov et al., 2014):

$$
\left.\begin{aligned}
T_{Br}^{v} &= \sum_{j=1}^{2} \frac{T_j |W_j|^2}{\left| 1 - r_j^- r_j^+ \exp(2i\psi_j) \right|^2} \times [\left(1 - \exp\left(-2\,\mathrm{Im}\,\psi_j\right)\right)\left(1 + \left| r_j^- \exp\left(i\psi_j\right) \right|^2\right) + \\
&\quad + 4\frac{\mathrm{Im}\,\psi_j}{\mathrm{Re}\,\psi_j}\mathrm{Re}\left(r_j^- \exp\left(i\psi_j\right)\right)\mathrm{Im}\left(\exp\left(i\psi_j\right)\right) \times \left(\frac{\left|k_{z_j}\right|^2 - k_x^2}{\left|k_j\right|^2}\right)]\frac{\mathrm{Re}\,Z_j}{\mathrm{Re}\,Z_0} + T_3 |W_3|^2 \frac{\mathrm{Re}\,Z_3}{\mathrm{Re}\,Z_0}, \\
T_{Br}^{h} &= \sum_{j=1}^{2} \frac{T_j |W_j|^2}{\left| 1 - r_j^- r_j^+ \exp(2i\psi_j) \right|^2} \times [\left(1 - \exp\left(-2\,\mathrm{Im}\,\psi_j\right)\right)\left(1 + \left| r_j^- \exp\left(i\psi_j\right) \right|^2\right) + \\
&\quad + 4\frac{\mathrm{Im}\,\psi_j}{\mathrm{Re}\,\psi_j}\mathrm{Re}\left(r_j^- \exp\left(i\psi_j\right)\right)\mathrm{Im}\left(\exp\left(i\psi_j\right)\right)]\frac{\mathrm{Re}\,Z_j}{\mathrm{Re}\,Z_0} + T_3 |W_3|^2 \frac{\mathrm{Re}\,Z_3}{\mathrm{Re}\,Z_0},
\end{aligned}\right\} \tag{1}
$$

where the superscripts at $T_{Br}$ denote: $v$ − V-pol and $h$ − H-pol; subscript $j$ equal to: 0 denotes free space (atmosphere), 1 snow cover, 2 lake ice, 3 water; $k_j$ is wave vector in layer $j$; $T_j$ is physical temperature of layer $j$; $r_j^+$, $r_j^-$ are reflection coefficient at the upper and bottom boundary of layer $j$, respectively; $\psi_j$ is total retarding phase across layer $j$; $W_j$ is coefficient of transmission from the inner side of the upper boundary of layer $j$ to the boundary of the layered structure (snow-atmosphere boundary in our case); $Z_j$ is characteristic impedance of layer $j$. The properties of each layer are defined by the following relations:

$$
\left.
\begin{aligned}
&W_j = \prod_{m=1}^{j} \frac{Z_{m-1}^{in+} + Z_{m-1}}{Z_{m-1}^{in+} + Z_m} \exp\left(i\psi_{m-1}\right), \quad Z_j^{in+} = Z_j \frac{Z_{j-1}^{in+} - iZ_j \operatorname{tg}\psi_j}{Z_j - iZ_{j-1}^{in+}\operatorname{tg}\psi_j}, \quad Z_j^{in-} = Z_j \frac{Z_{j+1}^{in-} - iZ_j \operatorname{tg}\psi_j}{Z_j - iZ_{j+1}^{in-}\operatorname{tg}\psi_j}, \\[2mm]
&\psi_j = k_{z_j} h_j, \quad k_{z_j} = k_j \cos\theta_j, \quad k_x = k_0 \sin\theta_0, \quad k_j = \frac{2\pi\sqrt{\varepsilon_j}}{\lambda}, \quad \cos\theta_j = \sqrt{\frac{\varepsilon_j - \varepsilon_0 \sin^2\theta_0}{\varepsilon_j}}, \\[2mm]
&r_j^+ = \frac{Z_{j-1}^{in+} - Z_j}{Z_{j-1}^{in+} + Z_j}, \quad r_j^- = \frac{Z_{j+1}^{in-} - Z_j}{Z_{j+1}^{in-} + Z_j}, \quad Z_j = \begin{cases} \dfrac{1}{\sqrt{\varepsilon_j}\,\cos\theta_j} & -\text{ horizontal polarization} \\[3mm] \dfrac{\cos\theta_j}{\sqrt{\varepsilon_j}} & -\text{ vertical polarization} \end{cases},
\end{aligned}
\right\} \tag{2}
$$

where $Z_j^{in+}$ is input impedance in layer $j$ at the bottom boundary for a wave propagating upward; $Z_j^{in-}$ is input impedance in layer $j$ at the upper boundary for a wave propagating downward; $h_j$ is thickness of layer $j$; $\lambda$ is radiation wave length; $\theta_0$ is viewing angle (Fig. 6); $\varepsilon_j$ is complex dielectric constant of layer $j$. Therefore, the system radiation is determined by physical temperature and complex dielectric constant of each layer. Dielectric properties of a layer depend on its composition, temperature, wetness and other physical and structure characteristics. Besides, when the radiation wave length is comparable with the size of inhomogeneities in the medium, scattering effects should also be taken into account (Ishimaru, 1978; Bohren and Huffman, 1983). Such effects result in considerable changes in snow and ice dielectric and radiative properties (Boyarskii et al., 1994; Boyarskii and Tikhonov, 2000; Tikhonov et al., 2014). For ice and snow, such inhomogenieties can be ice grains, air pores and water droplets.

To determine dielectric properties of lake snow and ice, the quasi-wave model of effective dielectric constant $\varepsilon_{eff}$ of a multi-phase disperse medium (QWM) was employed (Boyarskii et al., 1994, 2002). According to the model, snow cover is represented by an air medium with inclusions of spherical ice grains coated in water film and water droplets (Fig. 7a), and lake ice by a continuous freshwater ice medium with spherical inclusions of air bubbles and water droplets (Fig. 7b). The effective dielectric constant of lake snow is derived from:

$$\varepsilon_{eff}^{snow} = \varepsilon_a \left[ 1 - \frac{4\pi n_i \left( \varepsilon_i^w + 2\varepsilon_a \right) \frac{\langle f_\omega \rangle_i}{k^2}}{2\varepsilon_{eff} + \varepsilon_i^w} - \frac{4\pi n_w \left( \varepsilon_w + 2\varepsilon_a \right) \frac{\langle f_\omega \rangle_w}{k^2}}{2\varepsilon_{eff} + \varepsilon_w} \right]^{-1}, \tag{3}$$

where $k$ is wave number; $n_i$ and $n_w$ are concentrations of ice grains and water droplets, respectively; $\varepsilon_a$ and $\varepsilon_w$ are complex dielectric constants of air and water, respectively; $\varepsilon_i^{\,w}$ is effective dielectric constant of an ice grain coated in water film (Boyarskii et al., 2002); $\langle f_\omega \rangle_i$ and $\langle f_\omega \rangle_w$ are size-averaged amplitudes of forward scattering on an ice grain coated in water film and a water droplet, respectively. The effective dielectric constant of lake ice is derived from:

$$\varepsilon_{eff}^{lake\ ice} = \varepsilon_i \left[ 1 - \frac{4\pi n_a \left( \varepsilon_a + 2\varepsilon_i \right) \frac{\langle f_\omega \rangle_a}{k^2}}{2\varepsilon_{eff} + \varepsilon_a} - \frac{4\pi n_w \left( \varepsilon_w + 2\varepsilon_i \right) \frac{\langle f_\omega \rangle_w}{k^2}}{2\varepsilon_{eff} + \varepsilon_w} \right]^{-1}, \tag{4}$$

where $n_a$ is concentration of air bubbles; $\varepsilon_i$ is complex dielectric constant of ice; $\langle f_\omega \rangle_a$ is size-averaged amplitude of forward scattering on air bubbles. The amplitudes of forward scattering $f_\omega$ in Eqs. (3) and (4) are calculated by the Mie theory (Bohren and Huffman, 1983). The triangle parentheses indicate averaging over the ensemble of scatterers. Sizes of the inclusions in lake ice and snow cover are assumed to obey lognormal distribution (Colbeck, 1987). The model considers both physical and structural properties of lake ice and snow, as well as scattering effects due to structural inhomogeneities

of the media. Absorption and scattering on particle inhomogeinities (spherical ice grains, air bubbles, water droplets, ice grains coated in water) are represented in Eqs. (3) and (4) by term $f_\omega$ which is the amplitude of forward scattering on a spherical particle. This is substantiated by the forward scattering theorem stating that the extinction cross section determining radiation losses due to wave absorption and scattering on a particle depends only on the scattering amplitude in forward direction $f_\omega$ (Ishimaru, 1978). Also, note that the form of the inhomogeinities in real lake snow and ice, as a rule, is not strictly spherical (Gray and Male, 1986; Voitkovskiy, 1999; Cuffey and Paterson, 2010; Singh et al., 2011). However, as shown in (Bohren and Huffman, 1983), in the vicinity of the forward direction, non-spherical particles scatter radiation very similar to the way spherical ones of the same section do. So, the consideration of particle non-sphericity is not a principal issue for the model, it only complicates the calculations.

If the size of medium inhomogeneities is much less than the wavelength ($k_j \to 0$), Eqs. (3) and (4) transform into the mixing formulas of Polder-van Santen model (PVSM) (Boyarskii et al., 1994). According Polder and van Santen (1946), Eqs. (3) and (4) take the forms:

$$\varepsilon_{eff}^{snow} = \varepsilon_a \left[ 1 - \frac{3V_i \left( \varepsilon_i - \varepsilon_a \right)}{2\varepsilon_{eff} + \varepsilon_i} - \frac{3V_w \left( \varepsilon_w - \varepsilon_a \right)}{2\varepsilon_{eff} + \varepsilon_w} \right]^{-1}, \qquad (5)$$

$$\varepsilon_{eff}^{lake\,ice} = \varepsilon_i \left[ 1 - \frac{3V_a \left( \varepsilon_a - \varepsilon_i \right)}{2\varepsilon_{eff} + \varepsilon_a} - \frac{3V_w \left( \varepsilon_w - \varepsilon_i \right)}{2\varepsilon_{eff} + \varepsilon_w} \right]^{-1}, \qquad (6)$$

where $V_i$, $V_w$ and $V_a$ are the volumetric parts of the mixture components – ice, water and air.

Therefore, the complex dielectric constants $\varepsilon_0$ and $\varepsilon_j$ in Eq. (2) are defined as:

$$\varepsilon_0 = 1, \ \varepsilon_1 = \varepsilon_{eff}^{snow}, \ \varepsilon_2 = \varepsilon_{eff}^{lake\,ice}, \ \varepsilon_3 = \varepsilon_w. \qquad (7)$$

Which pair of Eqs. (3) and (4) or Eqs. (5) and (6) should be used to calculate $\varepsilon_1$ and $\varepsilon_2$ depends on the ratio of the radiation wave length and the mean size of the medium (snow or lake ice) inhomogeineities.

Scattering on a rough surface was taken into account using the model described in (Choudhury et al., 1979), according to which the snow-atmosphere boundary reflection coefficients are multiplied by a factor (Tikhonov et al., 2014):

$$\left(\exp\left(-l\cos^2\theta_0\right)\right)^{1/2}.$$

(8)

The roughness parameter $l$ is defined as:

$$l = 4\sigma^2\left(\frac{2\pi}{\lambda}\right)^2,$$

(9)

where $\sigma$ is surface roughness standard deviation.

### 3.2    Model modification

To analyze the radiative properties of the study lakes in various seasons, snow cover and lake ice

radiation penetration depth dependencies on wavelength of electromagnetic radiation were calculated using QWM and PVSM. Penetration depth is the thickness of a medium layer at which the amplitude of an electromagnetic wave penetrating the medium drops $e$ times. Penetration depth is considered to define the thickness of effectively radiating layer of the medium that can be calculated as (Sharkov, 2003; Kang, 2012):

$$h_{PD} = \frac{\lambda}{4\pi\,\mathrm{Im}\left(\sqrt{\varepsilon_j}\right)}.$$

(10)

Snow cover and lake ice penetration depths were determined for cold (-20$^\circ$ C) and warm (0$^\circ$ C)

conditions. At -20$^\circ$ C, snow is dry and lake ice contains minimal quantity of liquid water at its bottom in contact with lake water with temperature close to 0$^\circ$ C. At 0$^\circ$ C, intensive melt process sets on. In this case, the maximum volumetric wetness of snow cover (volume of water contained in unit volume of

snow) is determined by its density and structure and ranges 9-30%, but mostly 12-15% (Kuz'min, 1957;

Kotlyakov, 2000). Excess snow water trickles down to ice surface, and then leaks into cracks in ice increasing its wetness. During the melt period, wetness of lake ice is the highest and depends on its porosity, that is the volumetric fraction of cavities (pores, capillaries, cracks) that can be filled with water. To calculate the penetration depth, we used real physical parameters of lake ice and snow cover

(Kuz'min, 1957; Gray and Male, 1986; Voitkovskiy, 1999; Kotlyakov, 2000; Cuffey and Paterson, 2010; Singh et al., 2011) presented in Table 2. Results of the calculations are shown in Fig. 8. Let us consider the results obtained for MIRAS (1.4 GHz). As follows from Fig. 8, the penetration depth of dry snow is of the order of 1 km. Under negative temperatures, the penetration depth of lake ice is also large and equals several meters. Note that for both media, the calculations by QWM and PVSM are the same.

The fact is explained by the negligible effects of scattering on ice grains and air bubbles at this wave frequency (Tikhonov et al., 2014). Under negative temperatures, the effective dielectric constants of snow and lake ice can be calculated by Eqs. (5) and (6). The differences in penetration depths of continuous freshwater ice and lake ice are caused by the presence of a small amount of liquid water in lake ice (Table 2) which absorbs radiation so that the penetration depth diminishes. Around $0^{\circ}$ C, lake

ice penetration depth falls significantly, to about 40 cm (Fig. 8). Such a dramatic change is associated with liquid water accumulating in lake ice (Table 2) and, consequently, growing radiation absorption. Here, the scattering effects are also negligible (Tikhonov et al., 2014) and Eq. (6) can be used to calculate the effective dielectric constant. For wet snow, however, the effect of scattering on water droplets and ice grains coated in water is significant (Tikhonov et al., 2014). This explains the

difference in penetration depths estimated by QWM and PVSM (Fig. 8). Therefore, in presence of liquid water, the effective dielectric constant of snow cover should be obtained by Eq. (3). So, given the thicknesses of snow cover and lake ice of the study lakes (Fig. 4), these media will be almost completely transparent for the 1.4 GHz radiation under negative temperatures. At the same frequency, with the appearance of liquid water, the penetration depth of snow falls to only a few centimeters, that

of lake ice a few dozens of centimeters (Fig. 8), and snow and ice covers of the lakes become non-transparent for the radiation.

Atmospheric radiation and absorption of radiation by the atmosphere were taken into account in the calculations of brightness temperature. For this purpose, a model of standard atmosphere detailed in

(Tikhonov et al., 2014) was used. Generally, brightness temperature $T_{br\,syst}^{h,v}$ of the water surface – lake ice – snow cover – atmosphere system is determined as (Tikhonov et al., 2014):

$$T_{br\,syst}^{h,v} = T_{Br}^{h,v} \exp\left(-\xi_{\theta_0}\right) + T_a + R^{h,v} T_a \exp\left(-\xi_{\theta_0}\right) + R^{h,v} T_{\cos} \exp\left(-2\xi_{\theta_0}\right), \tag{11}$$

where $T_{Br}^{h,v}$ is calculated by Eqs. (1) and (2), either for ice-free water surface or for lake ice with snow cover; $T_a$ is brightness temperature of the atmosphere; $T_{cos}$ is brightness temperature of cosmic radiation; $R^{h,v} = (r_1^+)^2$ is reflectivity of either water surface or lake ice with snow cover; $\xi_{\theta_0}$ is atmospheric absorption depending on viewing angle $\theta_0$. The determination of $T_a$, $T_{cos}$ and $\xi_{\theta_0}$ is detailed in (Tikhonov et al., 2014).

Equations (11), (1) and (2) were used to calculate seasonal brightness temperature variations of each of the study lakes. The calculations were made for every day during the whole study period (2012-2013). Input parameters were real meteorological and glaciological data of the lakes and their regions averaged over a few decades (Figs. 2-4). The thickness of ice cannot be the same across the whole lake area, it depends on many factors (depth, underwater currents, air temperature, etc.). So, in the calculations, brightness temperature was averaged over ice depth within $\pm 10$ cm around the mean current value (Fig. 4). The effective temperature of a lake ice layer was assumed equal to the mean of air temperature and water temperature beneath ice layer, but not higher $0^\circ$C.

It was also assumed that $\sigma$ in Eq. (9) amounted to ~2 cm that corresponds to the mean roughness of snow cover (Rees, 2006).

In the modeling, we used mean seasonal values of snow density, ice grain diameter, volumetric wetness of snow, ice porosity and size of air bubbles in ice. These characteristics were taken from various sources (Kuz'min, 1957; Gray and Male, 1986; Kotlyakov, 2000; Kozlov, 2000; Cuffey and Paterson, 2010; Barry and Gan, 2011; Singh et al., 2011). It was assumed that in the transitional period (from TR2 to TR3), a gradual increase in lake ice porosity, as well as snow and ice wetness took place. These parameters are listed in Table 3. They were almost the same for all the lakes.

## 4. Results and discussion

Results of the calculations are presented in Fig. 5 (thick lines). From the figure, we can see that the model calculations agree well with satellite data across all five lakes. From model calculations and analysis of the results, the following features typical of TR1, TR2 and TR3 for the study lakes can be inferred.

TR1 (Fig. 5) is associated with water surface clear of ice and spans: mid May – mid December for Baikal, May – mid January for Ladoga, July – end of November for GBL, and beginning of June - end of November of GSL (Fig. 5). In TR1, brightness temperature is determined by water surface temperature and waves. From September to December, it is the season of high winds on Ladoga. Wind speed often reaches 15-25 m/s and waves 5-8 m in height. Sometimes, the water surface appears quite chaotic: waves get jumbled up with high and steep ones propagating in various directions (Rumyantsev, 2015). On Baikal, the season of high winds begins in September - October and lasts until complete freeze over. During this period, waves reach 5-6 meters and more in height (Galaziya, 1984; Tulokhonov, 2008). This peculiarity explains Ladoga and Baikal brightness temperature oscillations (by MIRAS data) demonstrated in Fig. 5. On GBL and GSL, winds hardly ever exceed 10-12 m/s (Woo et al., 2007), which determines small amplitude oscillations of brightness temperatures (by MIRAS data) for these lakes (Fig. 5). Note that wind waves at water surface clear of ice are not taken into account in the model calculations. So, the model brightness temperature curve appears smooth since it is defined only by the temperature of water surface layer (Fig. 3). Huron almost never freezes over completely (Rumyantsev et al., 2012). In 2012-2013, it remained practically clear of ice, with the exception of the bays and coastline (see references to the NOAA website above). Therefore, TR1 spans over the whole duration of Huron observation, which is confirmed by model calculations (Fig. 5). Here, brightness temperature variations are determined by water surface temperature and waves. The latter depend on wind speed that did not exceed 10 m/s during the observation (see references to the NOAA website above).

TR2 brightness temperature in Fig. 5 is associated with complete ice cover. For Baikal, it lasts from mid-December to April; for Ladoga, from February to April; for GBL and GSL, from mid-November to May. In the beginning of TR2, there is a sharp rise in brightness temperature by 20-60 K,

depending on polarization and meteorological conditions. It is associated with the initiation of complete freeze over. After that, brightness temperature gradually increases by about 10 K as lake ice thickness grows (Kang et al., 2010, 2014). It can reach 100-140 cm on Baikal, 40-60 cm on Ladoga. 100-120 cm on GBL and GSL (Galaziya, 1984; Tulokhonov, 2008; Kang et al., 2010, 2012; Rumyantsev et al.,

2012; Rumyantsev, 2015; https://www.polardata.ca/pdcsearch/). Brightness temperature oscillations (by MIRAS data) in TR2 are explained by the interference of electromagnetic waves occurring due to reflection at the upper and bottom boundaries of the ice cover. During this period, ice features low dielectric losses, because it has small quantity of pores (cracks) and little liquid water. The penetration depth at the 21 cm wave length is much greater than the thickness of ice (Fig. 8), so the ice cover

represents an added radiative layer to the water surface. This fact is fully supported by the model calculations (Fig. 5). Presence of dry snow on ice has almost no effect on radiative properties of the lakes, because dry snow is practically radio transparent at the given frequency (1.4 GHz) (Fig. 8). Also, brightness temperature oscillations (by MIRAS data) in TR2 can be related to local temperature variations with short-lasting snow melts due to solar radiation followed by freezing.

15        TR3 in Fig. 5 features short-lasting abrupt rise in brightness temperature by 40-90 K. It is the period of dramatic change in ice cover structure (intense deterioration and melt) and snow melt. Brightness temperature is affected by rapidly changing physical properties of ice: opening of numerous cracks that are filled with liquid water. This changes the dielectric properties of ice inducing strong rise in the absorption of electromagnetic radiation. Higher absorption in turn leads to ice brightness

temperature growth, since the penetration depth dramatically falls (to less than half ice thickness, Fig. 8) which results in obturation of microwave radiation from the water surface. In TR3, the snow cover is thin (5-10 cm) and wet (4-16 %) (Table 3, Fig. 4). The snow cover penetration depth is not large (~20 cm), although larger than the thickness of snow on lake ice (Table 3, Figs. 4, 8). So, the layer of wet snow has the same impact on radiation in TR3 as ice cover does in TR2. In TR3, snow cover provides

additional radiation to that of melting ice. Figure 9 confirms this conclusion. The figure presents modeling results of seasonal brightness temperature variations at Baikal in 2012 for two media versions in TR2 and TR3: ice cover with and without snow layer above. Model input parameters are listed in Table 3. According to Fig. 9, in TR2 snow cover has virtually no impact on brightness temperature. The

snow is dry and radio transparent at 1.4 GHz. In TR3, the main contribution to total radiation is provided by the deteriorating wet ice cover, although the contribution of snow is significant. This is linked to the appearance of liquid water and, consequently, growing effects of scattering and absorption in the snow. Note that the input of wet snow to total radiation is bigger for H-pol than it is for V-pol.

TR3 lasts from the beginning of April to May at Baikal, from May to July at GBL, throughout May at GSL, and for two weeks in April at Ladoga (Fig. 5). The calendar limits of TR3 vary considerably because of varying meteorological conditions of the study lakes.

Note that the beginning of TR1 corresponds not to the time of complete clearing of lake ice, but rather the moment when lake ice concentration gets well below 100% and snow-melt puddles appear on

its surface. At 1.4 GHz, water penetration depth is 2-3 cm (Galakhov, 1980). So, even a thin layer of water (~3 cm) is imaged by the radiometer quite like free water surface.

Results of the present work agree with those published by Kang (2012) and Kang et al. (2012). In these papers, the authors investigate ice cover phenology of GBL and GSL derived from AMSR-E data. Lake ice and snow penetration depths for AMSR-E frequencies (6.9, 18.7, 23.8, 36.5, 89.0 GHz)

are much less than for MIRAS (Fig. 8). At 6.9 GHz, the penetration depth is ~30 cm for dry lake ice and 0.5-2 cm for wet ice and wet snow (Fig. 8). Consequently, during the cold winter period, when almost no liquid water is present in snow and lake ice, AMSR-E receives electromagnetic radiation emitted by the upper layer (~30 cm) of the ice cover. In warm season, when snow and lake ice become wet, AMSR-E receives radiation only from the 0.5-2 cm thick surface layer. Therefore the effect of abrupt

brightness temperature rise during ice cover deterioration and melt period is much less pronounced for AMSR-E than MIRAS. AMSR-E can record only snow or upper ice layer melt, since it receives radiation practically from the surface. In contrast, MIRAS, even in warm season, receives radiation from substantially larger depths (~20-40 cm), thus recording changes within the ice cover (deterioration and melt).

It is noteworthy, that the above analysis of seasonal changes of brightness temperature of the lakes was based on theoretical modeling and model calculations. As input model parameters we took mean seasonal characteristics  of snow cover and lake ice (density, ice grain size, porosity, etc.) as well as multi-year averages of meteorological parameters (air and water temperature, ice and snow

thickness). Such parameters as ice and snow density, ice grain and air pore size have little effect on lake ice brightness temperature due to absence of scattering and low absorption at L-band (see Section 3.2 and Fig. 8). However, air, snow and ice temperature and especially volumetric wetness of snow and ice are the key factors in calculating brightness temperature of the water surface – lake ice – snow cover – atmosphere system. This is clearly demonstrated in Fig. 5 showing slight discrepancies between model results and satellite data in various years. These discrepancies are caused by divergence of actual parameters (snow and ice wetness, air, snow and ice temperature) from their averages.

## 5. Conclusions

Using the authors' model of thermal emission of multi-layered non-isothermal media, an analysis of ice cover phenology phases of large freshwater bodies based on SMOS MIRAS data (1.4 GHz) was performed by examples of lakes Baikal, Ladoga, GBL, GSL, and Huron. The employed frequency range (L-band) allowed highlighting the state of ice and snow at greater depths compared to frequency ranges of AMSR-E and AMSR2. The theoretical analysis showed that L-band thermal emission was received practically from the whole lake ice column during both cold winter and melt periods. This makes it possible to remotely sense processes in ice and snow cover even in the melt period. For the first time SMOS MIRAS data were used to assess the phenology phases of freshwater lakes. Before, AMSR-E and AMSR2 with higher operational frequencies had been used for the purpose in the melt period providing information only about the surface layer of lake ice. Our study not only replicated earlier results for AMSR-E and AMSR2 (see references in Introduction), but also complemented them with new findings due to greater radiation penetration depth at MIRAS frequency (1.4 GHz).

Comparison of the satellite data (SMOS MIRAS) and modeling results revealed three distinct brightness temperature time regions at the freezing lakes that are associated with: water surface clear of ice (TR1); complete ice cover (TR2); short-lasting abrupt rise in brightness temperature by 40-90 K marking the period of dramatic change in ice cover structure - intense deterioration and melt (TR3).

The revealed patterns of seasonal variations of brightness temperature and related lake ice phonological phases can be used to assess the hydrological regime of freezing inland water bodies

(rivers, lakes, water reservoirs) based on satellite microwave radiometry data. The obtained results can be useful for setting the operating terms of winter crossings and roads on ice, since with the beginning of ice deterioration, these transportation routes across water bodies become insecure and cannot be used any more.

*Author contribution.* The present work was guided by Prof. Evgeniy Sharkov. Model development and calculations were conducted by Dr. Vasiliy Tikhonov and Prof. Evgeniy Sharkov. Acquisition, processing and assimilation of the satellite data were performed by Dr. Ilya Khvostov. Dr. Andrey Romanov analyzed the satellite data, as well as climatic, meteorological and glaciological characteristics of the study lakes and regions. All authors took part in the discussions of the received
results. The preparation of the manuscript was the responsibility of Dr. Vasiliy Tikhonov.

*Competing interests.* The authors declare that they have no conflict of interest.

*Acknowledgements.* The SMOS data used in this study were obtained under the European Space Agency Project No. 4747 "Remote mapping of Siberian saline soils". The work was supported by FASO Russia (Theme "Monitoring", State Register No. 01.20.0.2.00164). The study of snow cover and
15 its influence on microwave radiation of freezing lakes was partially supported by the Russian Foundation for Basic Research (RFBR grants No. 18-05-00427, 18-05-00753).

We thank T.Yu. Bocharova for translating this paper.

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

**Table 1.**
Study lakes: locations, center coordinates of examined SMOS L1C grid cells.

| Lake Name | Region | Latitude | Longitude |
|---|---|---|---|
| Baikal | Southern Siberia, Russia | $54.17^\circ$ N | $108.91^\circ$ E |
| Ladoga | Northwestern Russia | $60.88^\circ$ N | $31.37^\circ$ E |
| Huron | Great Lakes of North America | $44.56^\circ$ N | $82.41^\circ$ W |
| Great Slave | Northwest Territories of Canada | $61.28^\circ$ N | $114.80^\circ$ W |
| Great Bear | Northwest Territories of Canada | $65.97^\circ$ N | $120.57^\circ$ W |

**Table 2.**
Properties of snow cover and lake ice taken for penetration depth calculations. Standard deviations of ice grain and air bubble sizes were assumed at 40 %.

|  | t=-20° C | t=0° C |
|---|---|---|
| **Snow cover** | | |
| dry density | 0.3 g/cm$^3$ | 0.3 g/cm$^3$ |
| diameter ice grain | 0.04 cm | 0.06 cm |
| volumetric wetness | 0 % | 10 % |
| **Lake ice** | | |
| porosity | 12 % | 50 % |
| diameter air bubble | 0.1 cm | 0.1 cm |
| volumetric wetness | 3 % | 25 % |

**Table 3.**
Properties of snow cover, lake ice and water taken for brightness temperature modeling in TR1, TR2 and TR3. Standard deviations of ice grain and air bubble sizes were assumed at 40 %.

| Lacustrine Environment | TR1 | TR2 | TR3 |
|---|---|---|---|
| **Water** | | | |
| Temperature | $T_{Water}$, (Fig. 3) | $T_{Water}$, (Fig. 3) | $T_{Water}$, (Fig. 3) |
| **Lake ice** | | | |
| Temperature | - | $(T_{Water} + T_{Air}) / 2 \leq 0$ | $0^{\circ}$ C |
| Porosity | - | 10-15 % | 50 % |
| Diameter air bubble | - | 0.1 cm | 0.1 cm |
| Volumetric wetness | - | 3-8 % | 25-50 % |
| Thickness | - | $H_{ice}$, (Fig. 4) | $H_{ice}$, (Fig. 4) |
| **Snow cover** | | | |
| Temperature | - | $T_{Air}$, (Fig. 2) | $0^{\circ}$ C |
| Dry density | - | 0.3 g/cm$^3$ | 0.3 g/cm$^3$ |
| Diameter ice grain | - | 0.04 cm | 0.06 cm |
| Volumetric wetness | - | 0 % | 4-16 % |
| Thickness | - | $H_{snow}$, (Fig. 4) | 5-10 cm |

**Figure Captions**

Figure 1. Maps of Baikal, Ladoga, GBL, GSL, and Huron. Green circles mark the SMOS L1C grid cells examined in the study.

Figure 2. Seasonal variations of air temperature in the study regions.

Figure 3. Seasonal variations of water surface temperature in the central parts of the study lakes.

Figure 4. Seasonal variations of ice thickness up to the maximum value and thickness of snow on ice.

Figure 5. Seasonal variations of brightness temperature within the SMOS L1C grid cells for Baikal, Ladoga, GBL, GSL, and Huron. Thin lines and symbols denote the SMOS data, thick lines represent the model results. Numbers mark brightness temperature TRs associated with: 1 - open water surface (TR1); 2 - complete ice cover (TR2); 3 - ice deterioration and melt (TR3) (see text).

Figure 6. Model of the "water – lake ice – snow cover – atmosphere" radiating system.

Figure 7. Models of snow cover (a) and lake ice (b).

Figure 8. Radiation penetration depth in continuous freshwater ice, snow and lake ice calculated by the PVSM and QWM models. Lake snow and ice parameters are given in Table 2. AMSR-E and MIRAS frequencies are indicated by vertical dashed lines.

Figure 9. Model variations of brightness temperature within the SMOS L1C grid cell for Baikal. TR2 and TR3 are calculated for ice without snow (rose and blue lines) and covered by snow (red and blue lines).

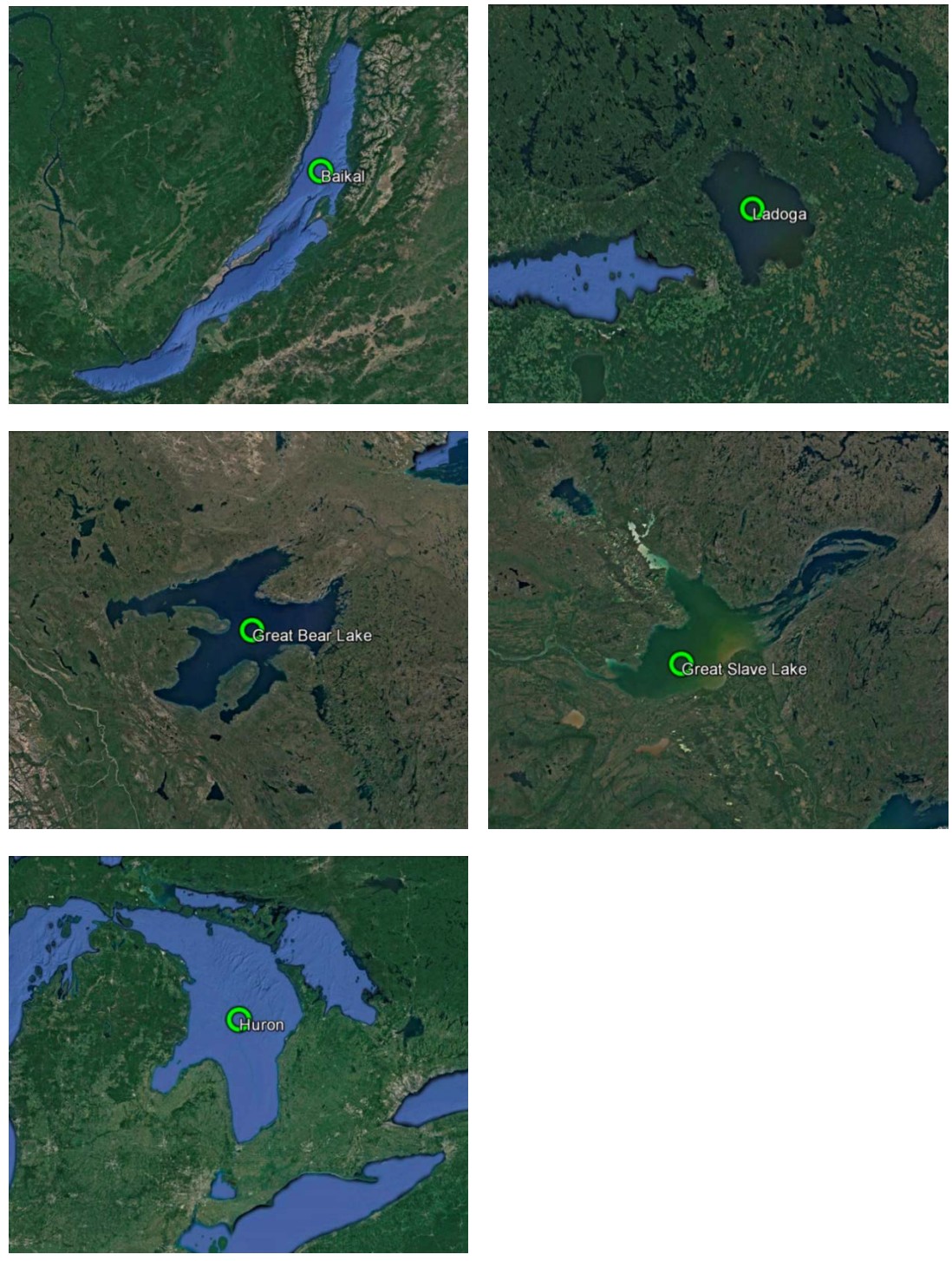

**Figure 1. Maps of Baikal, Ladoga, GBL, GSL, and Huron. Green circles mark the SMOS**
5  **L1C grid cells examined in the study.**

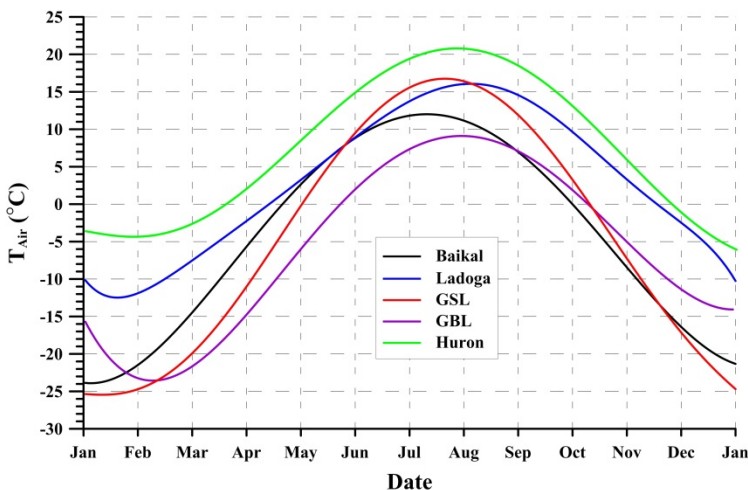

**Figure 2. Seasonal variations of air temperature in the study regions.**

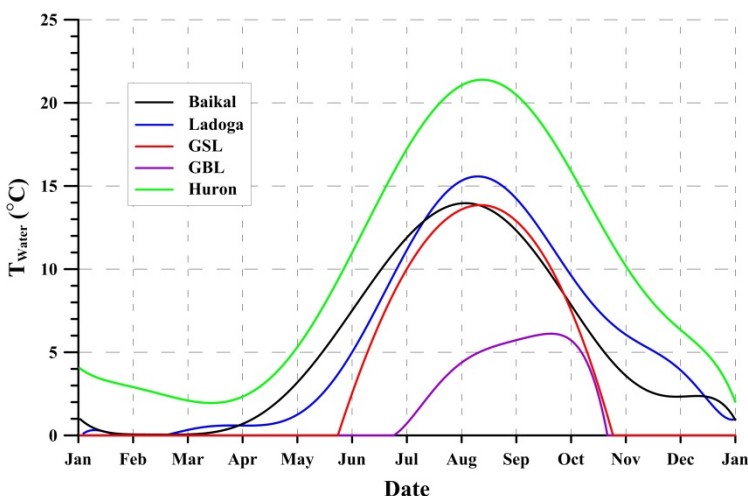

**Figure 3. Seasonal variations of water surface temperature in the central parts of the study lakes.**

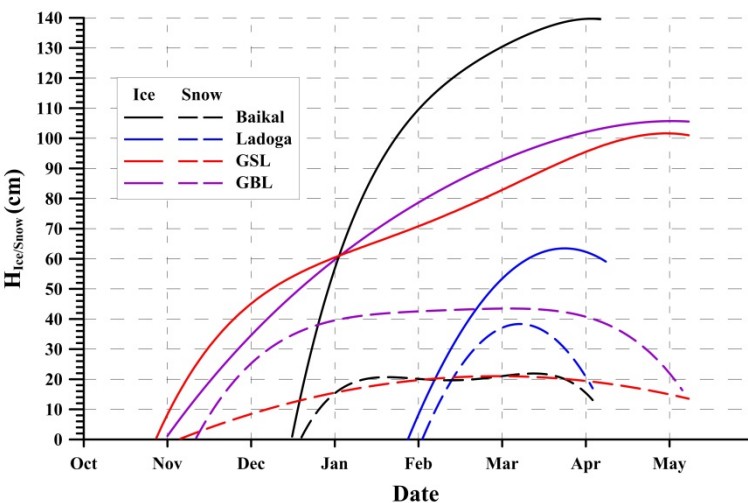

**Figure 4.** Seasonal variations of ice thickness up to the maximum value and thickness of snow on ice.

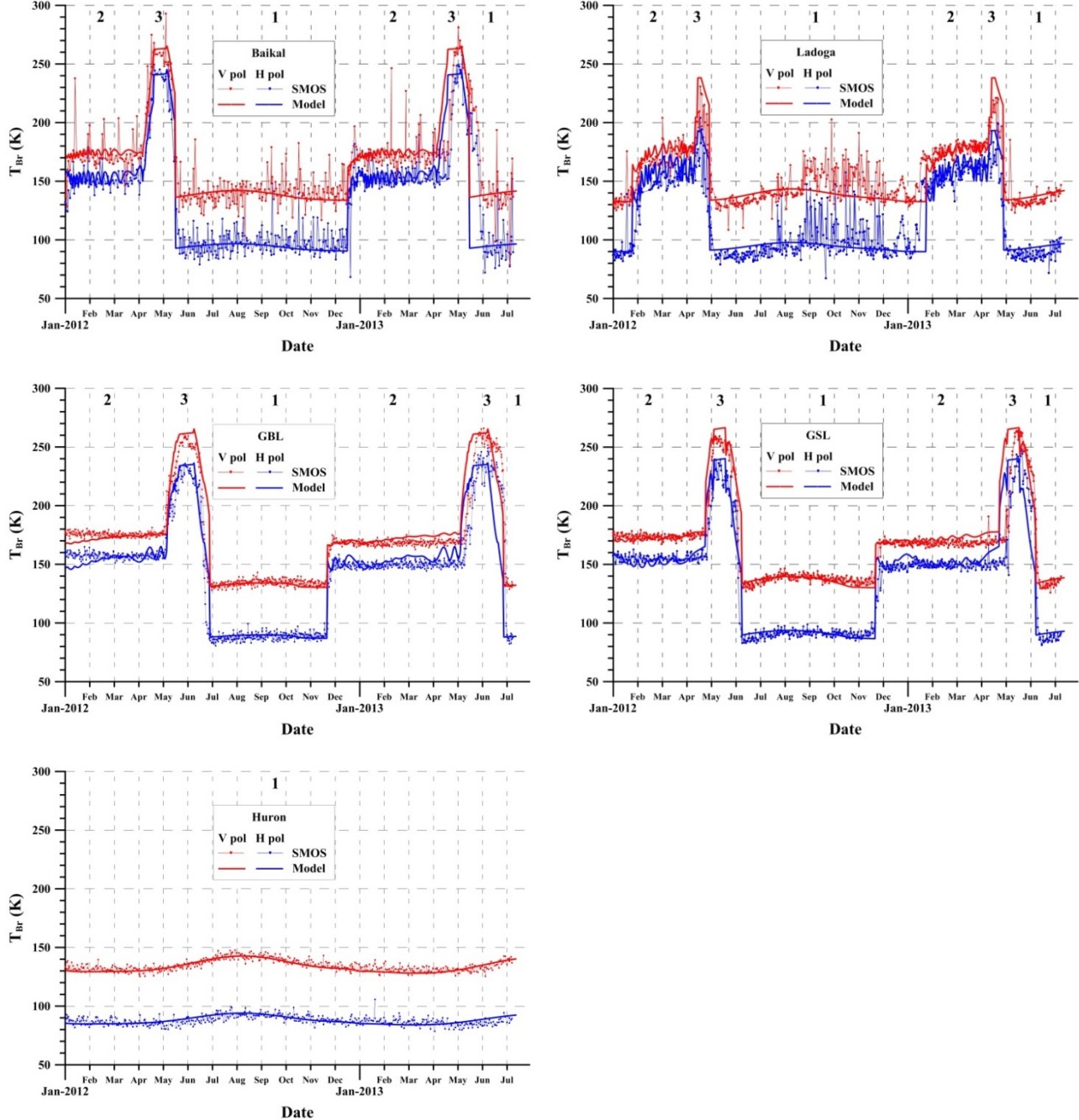

**Figure 5.** Seasonal variations of brightness temperature within the SMOS L1C grid cells for Baikal, Ladoga, GBL, GSL, and Huron. Thin lines and symbols denote the SMOS data, thick lines represent the model results. Numbers mark brightness temperature TRs associated with: 1 - open water surface (TR1); 2 - complete ice cover (TR2); 3 - ice deterioration and melt (TR3) (see text).

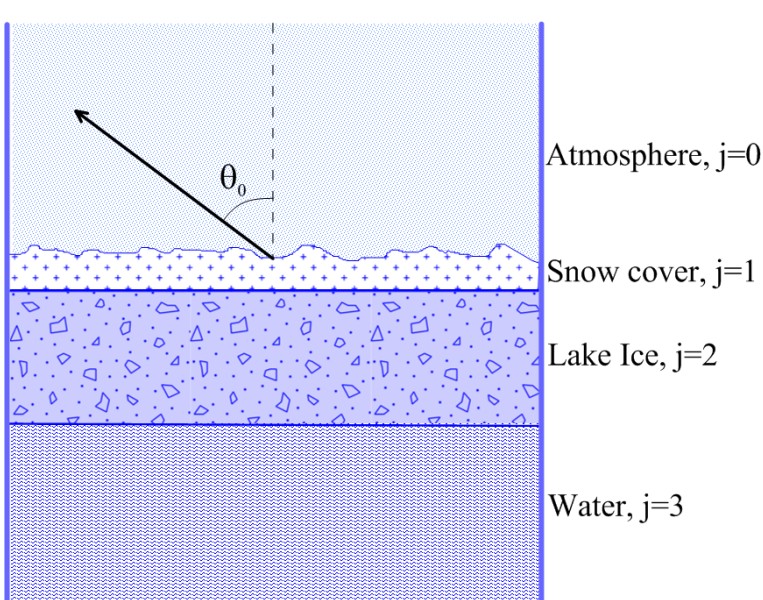

**Figure 6. Model of the "water – lake ice – snow cover – atmosphere" radiating system.**

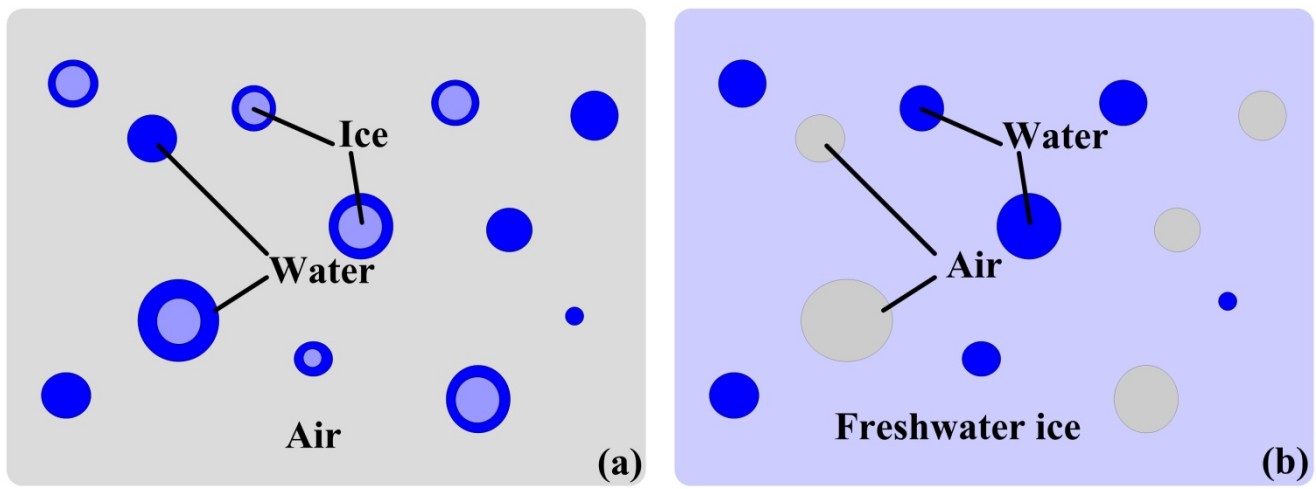

**Figure 7. Models of snow cover (a) and lake ice (b).**

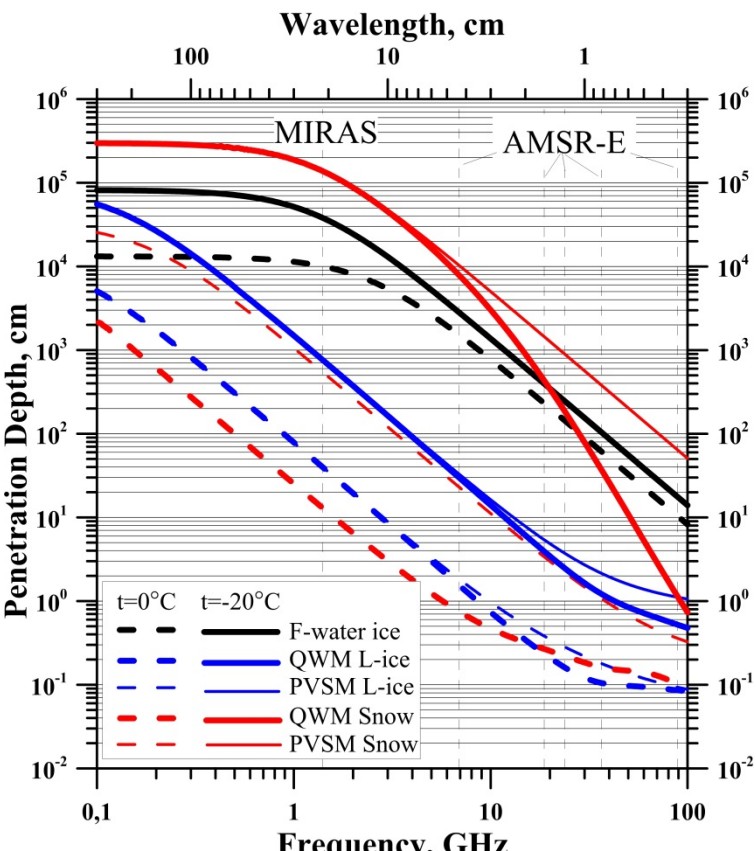

**Figure 8.** Radiation penetration depth in continuous freshwater ice, snow and lake ice calculated by the PVSM and QWM models. Lake snow and ice parameters are given in Table 2. AMSR-E and MIRAS frequencies are indicated by vertical dashed lines.

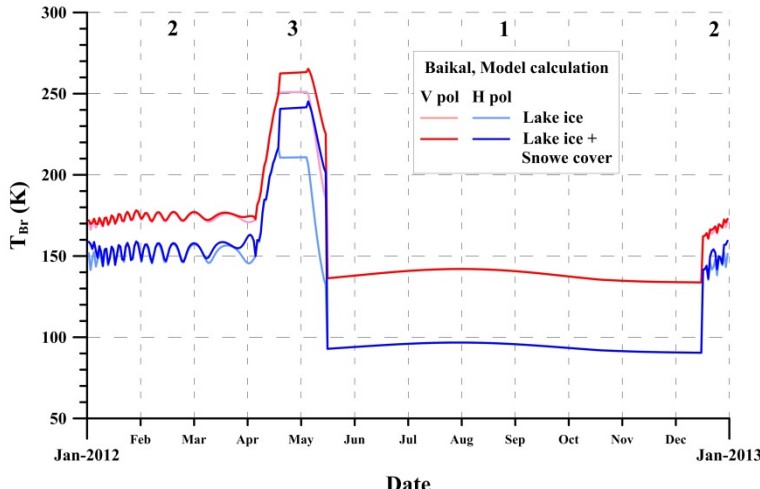

**Figure 9. Model variations of brightness temperature within the SMOS L1C grid cell for Baikal. TR2 and TR3 are calculated for ice without snow (rose and blue lines) and covered by snow (red and blue lines).**