# Peer review of "Theoretical study of ice cover phenology at large freshwater lakes based on SMOS MIRAS data"

_The Cryosphere, 2018_

## Referee Comment (RC1) · Anonymous Referee #1 · 8 Mar 2018

The authors studied L-band microwave emissions over large high-latitude lakes (except Huron) likely experiencing different lake ice phenology phases. The seasonal variations of SMOS brightness temperature were simulated and interpreted using a theoretical model developed previously for sea ice studies. The model-based approach is needed for lake ice studies and there are no major issues with the analysis; however, I do not find major contributions to the current understanding of remote sensing of lake ice either.

Major comments: (1) This is basically a theoretical study but no major improvements or new developments in the microwave emission modelling are found. Section 3 is based on the available literatures and has significant overlap in content with (Tikhonov et al., 2014) (e.g. Eq. 1 and 3, Fig. 7 are presented in both literatures). It is necessary to

review the theoretical basis, but this new study is not supposed to limit to a recap.

(2) Section 3: What about the incoherent scattering from a rough surface of wet snow? How do you consider layered snow and/or ice in the modelling? What about the temperature vertical gradient of snow and/or ice layer, which is not considered in the model or simulations?

(3) The explanations on the brightness temperature patterns of ice, dry snow and wet snow are reasonable as similarly described in other literatures (e.g. Ulaby et al., 1986). The study would be more interesting if more deepened analysis was added. For example, an additional analysis on the quantified differences between multi-frequency (e.g. SMOS vs AMSR) responses to typical lake ice conditions and the implications to lake ice remote sensing.

(4) Fig. 5 compares the simulated and observed SMOS brightness temperatures. Considering a large number of inputs of the theoretical model and limited knowledge of the ground truth (e.g. the snow and ice parameters), such simulations can only be used for qualitative purpose. It would be more rigid to have the model validated first using a controlled or field experiment with thorough measurements of the required parameters. It is also recommended to state the input parameter uncertainties and their impacts on model simulations.

Minor comments: Section 2.2: detailed descriptions of the data sets (e.g. specific parameters; measurement accuracy; spatial and temporal representativeness) used in this study are needed instead of a simple list of data sources.

---

## Referee Comment (RC2) · Anonymous Referee #2 · 14 Mar 2018

Review of paper tc-2018-27 Tikhonov et al. "Theoretical study of ice cover phenology at large freshwater lakes based on SMOS data"

General

The study deals with investigating L-band passive microwave emissions over freshwater lakes from a theoretical perspective, employing a forward model to simulate brightness temperatures which are then compared to observations from the SMOS mission.

The subject is interesting as relatively few studies have focused on modeling the passive microwave signatures from lakes. There are none that I am aware of that focus specifically on L-band. Although at present the coarse spatial resolution of spaceborne passive microwave sensors at L-band ultimately constrains their applicability to

a limited number of large lakes, the study is still relevant given the specific advantages of this frequency compared to higher bands. In this way, this study also has the potential of bringing new information to the ongoing discussion of an operational follow-on sensor to the SMOS and SMAP exploratory missions.

From a general viewpoint the study is well written, clear and easy to follow. However, insufficient detail is given on key factors regarding the model simulations; I am left wondering if some kind of tuning of the forward model has been done. If so this is fine but clear details should be given so other groups may seek to replicate the results. Furthermore, comparisons of model results to observational data are very qualitative, e.g. no numbers (bias, RMSE etc.) are given. These factors weaken the broader conclusions that can be drawn from this study.

An additional aspect that I feel the authors omit is employing the full potential of SMOS data by analyzing data from different incidence angles. This may have relevance especially concerning conclusions made about sensitivity to dry snow (see major comments below).

Last, the references should be expanded to acknowledge more modeling work done for lakes at other microwave bands and studies related to SMOS and the cryosphere. Authors now refer mainly to their own work (with the exception of papers by Kang et al., which are extensively cited). In light of the above the potential of the paper is not really achieved. I give several recommendations below to help make this happen; I would recommend the authors should carefully address these comments before publication.

Major comments

1. Introduction, line 23: Add appropriate reference to the SMOS mission, e.g. Kerr et al., 2010.

2. Introduction, throughout: The authors should refer more broadly to recent publications on applying L-band for the cryosphere. Studies on L-band signatures from the

Antarctica (Macelloni et al.,), sea ice (Kaleschke et al.) and snow cover (Schwank et al.) have been published in recent years.

3. Although not dealing with lakes as such, the study by Schwank et al. (2015) should be of particular relevance as they also present a model for snow cover at L-band which should at least be cited. I assume the model could also be easily applicable for lake ice. Similarly, there are some other works looking at lake ice signatures at AMSR frequencies beyond Kang et al., which could be added (see e.g. Kontu et al., 2014).

4. Section 2.1: The authors use only an incidence angle of 42.5 degrees from the L1C data (actually a collection of snapshots from 40 to 45 degrees). Why was this angle chosen over others (available from nadir to 70 degrees)?

5. Related to the above: Incidentally, SMOS incidence angels close to 41 degrees were noted by Schwank et al. (2015) to be insensitive to dry snow (whereas observations at other incidence angles were found to be sensitive to the presence of dry snow). Close to 41 degrees sensitivity to dry snow cover disappears at V-polarization, due to the opposing effects of impedance matching and refraction. Perhaps this is the cause the authors see no change in observed or modeled Tb with dry snow cover, as stated on p12, section 4? This should be discussed. Looking at other incidence angles beyond 42.5 degrees might shed some light here.

6. Section 3 is quite long, I suggest to divide into subsections. Section 3.1 could present the model and relevant equations, while section 3.2 could present the simulation setup (use of data etc) specific for this study (i.e. starting from p8, line 18).

7. Related to the above: p10 lines 10-17: This does not really deal with results, but should be moved from section 4 to section 3.

8. Section 3, section 4 and Table 3: Judging from figure 5 an almost perfect match of model vs. observations is achieved but it is not perfectly clear if e.g., tuning of model input parameters (given in Table 1) was required to achieve this. For TR1 no tuning is

possible from what I see, but what about volumetric wetness for ice in TR2 and snow and ice in TR3? A range is given in Table 3, but it is not clear if these were defined individually for each lake? If no tuning was required, the authors could also highlight this.

9. P10 lines 5-7: Dry snow and ice are transparent at L-band yes, but they still affect the emitted brightness temperature from the surface beneath though impedance matching and a change in the refraction angle. See Schwank et al., 2015. I imagine these results are also applicable to the water-ice-snow system.

10. P12, line 19-20: Again, the insensitivity to snow may be due to the choice of incidence angle (although at H-pol, something could perhaps be seen). I suggest the authors analyze the response vs. incidence angle e.g. for one lake comparing simulations with the TR1, TR2 and TR3 setting against SMOS observations (e.g., the average Tb for those periods at different incidence angles).

11. Section 5: The conclusions section is too short. The authors should provide a more complete assessment of their study, including the main results.

Minor/Editorial

1. P8, lines 8 and 19, maybe elsewhere: I think the convention is to write "wavelength" as one word (not wave length). Furthermore on line 19, wavelength is not a synonym to frequency, as it is now implied by the parenthesis. Please rephrase.

2. P9, line 17: "Note that for both media..." Sentence seems incomplete.

3. P12, line 13: "most wet". Unclear what is meant. Almost wet? Mostly wet?

4. P12 lime 14: Typo: capitalized 'M' in 'cm'

5. P13, line 7: "even in the cold season"

---

## Author Comment (AC1) · 3 May 2018

We thank Referee #1 for carefully reading our manuscript and the comments. We attempted to provide answers to all his/her questions. They are below in red.

The authors studied L-band microwave emissions over large high-latitude lakes (except Huron) likely experiencing different lake ice phenology phases. The seasonal variations of SMOS brightness temperature were simulated and interpreted using a theoretical model developed previously for sea ice studies. The model-based approach is needed for lake ice studies and there are no major issues with the analysis; however, I do not find major contributions to the current understanding of remote sensing of lake ice either.

The theoretical model we use for the interpretation of seasonal variations of freshwater lake ice brightness temperature was not developed specially for sea ice. It is a model of thermal emission of multi-layered non-isothermal media (Sharkov, 2003). We have added the reference to its original description. The model provides satisfactory results for microwave emission of sea ice, Antarctica ice sheet, and snow cover. The references to the corresponding papers (Tikhonov et al., 2008; Tikhonov et al., 2013; Tikhonov et al., 2014; Tikhonov et al., 2017) are added to "Introduction".

Our "contribution to the current understanding of remote sensing of lake ice" is detailed in Sections 3 and 4 where we discuss radiation penetration depth that is much higher at 1.4 GHz than at AMSR-E and AMSR2 frequencies of over 6.9 GHz. The frequency range we use allow understanding the state of ice and snow of greater thicknesses. In cold season, the radiation can be received from the entire ice column, and almost likewise in the melt period. Thus, the processes in ice and snow can be estimated using remote sensing techniques even during ice melt. For the first time the SMOS MIRAS data are used to assess the phonological phases of freezing freshwater lakes. Up to date, higher-frequency channels of AMSR-E and AMSR2 have been used for the purpose. We have mentioned this in the revised "Introduction".

Major comments:

(1) This is basically a theoretical study but no major improvements or new developments in the microwave emission modelling are found. Section 3 is based on the available literatures and has significant overlap in content with (Tikhonov et al., 2014) (e.g. Eq. 1 and 3, Fig. 7 are presented in both literatures). It is necessary to review the theoretical basis, but this new study is not supposed to limit to a recap.

We use our model that was previously presented and employed for theoretical description of microwave emission of sea ice, Antarctica ice sheet, and snow cover (see above). It is not the aim of this manuscript to present a new model of thermal emission of multi-layered non-isothermal media. We attempt and, in our view, succeed to demonstrate the utility of our model in assessing the state of lake ice throughout the seasons. So, the validity of our model is once again proven. A few key formulas and figures are reproduced in the manuscript from previous works only to make the model easier to understand for the reader.

(2) Section 3: What about the incoherent scattering from a rough surface of wet snow? How do you consider layered snow and/or ice in the modelling? What about the temperature vertical gradient of snow and/or ice layer, which is not considered in the model or simulations?

Certainly, snow cover surface roughness is considered in model calculations. For the purpose, the model of scattering on statistically rough surface (detailed in (Choudhury et al., 1979)) is used. The model suggests that there is no correlation between the amplitudes of the waves scattered by two points on the surface. This is justified because the ice and snow that cover the lake surface during the cold period are seasonal phenomena, unlike multi-year hummocked sea ice. We do not focus on the issue and give only a reference to the detailed description of the "layered surface – atmosphere" emission model (Tikhonov et al., 2014). However, since Referee #1 has raised the question, we add a short description of the way the scattering on rough snow cover is taken into account (Section 3.1) and the roughness characteristics used (Section 3.2). Note, that here we once again cannot help overlapping with our previous results (see Comment 1). As said already, ice and snow over the study lakes are seasonal, so they do not have any prominent layered structure and we may neglect it. The beginning of Section 3 reads: "…the radiative system consists of lake water surface covered with lake ice and then snow…". Figure 6 presents this schematically.

In Section 3.2 we mention, that, in the calculations, we use the so-called effective temperature of the ice cover equal to the mean of air temperature and water temperature beneath ice layer, but not higher $0^{\circ}$C. Such approach is widely used in modeling of emission of various layered media (atmosphere, snow cover, etc.) because it is impossible to properly consider all the media characteristics (see, for example, HUT model (Pulliainen et al., 1999)).

(3) The explanations on the brightness temperature patterns of ice, dry snow and wet snow are reasonable as similarly described in other literatures (e.g. Ulaby et al., 1986). The study would be more interesting if more deepened analysis was added. For example, an additional analysis on the quantified differences between multi-frequency (e.g. SMOS vs AMSR) responses to typical lake ice conditions and the implications to lake ice remote sensing.

Perhaps it skipped the attention of Referee #1 that the end of Section 4 is dedicated to the discussion and comparison of our results (SMOS) and similar results (Kang, 2012; Kang et al., 2012) obtained from AMSR data.

(4) Fig. 5 compares the simulated and observed SMOS brightness temperatures. Considering a large number of inputs of the theoretical model and limited knowledge of the ground truth (e.g. the snow and ice parameters), such simulations can only be used for qualitative purpose. It would be more rigid to have the model validated first using a controlled or field experiment with thorough measurements of the required parameters. It is also recommended to state the input parameter uncertainties and their impacts on model simulations.

The modeling results presented in the manuscript demonstrate that there exist not just qualitative but also quantitative agreement between theoretical calculations and satellite data. In Section 4, we discuss in detail the influence of thickness and wetness of ice and snow on brightness temperature of the system. Such studies were previously conducted for sea ice and snow cover (Tikhonov et al., 2008, 2013, 2014, 2017). The corresponding references are given in the manuscript. So, in our view, it is incorrect to say that "such simulations can only be used for qualitative purpose". All presented modeling results agree with satellite data for the whole study period and for all lakes. As to the influence of input parameters variation on modeling results, this

problem was investigated in (Tikhonov et al., 2014), where the influence of media inhomogeneities (ice grains, air pores, water droplets, etc.) on its scattering and absorption was examined. The aim of the present work is to explain dramatic and fast brightness temperature changes that are revealed from SMOS data over consecutive seasons at large freezing freshwater lakes. And we achieve it quite elaborately.

Minor comments: Section 2.2: detailed descriptions of the data sets (e.g. specific parameters; measurement accuracy; spatial and temporal representativeness) used in this study are needed instead of a simple list of data sources.

The data used in the modeling and presented in Section 2.2 (temperature, wind speed, snow and ice thickness, etc.) are taken from the web sites of well-known and reputable research institutions and belong to them. We only use these data. Their detailed discussion and estimation of measurement accuracy and spatial and temporal representativeness are outside the scope of our work and demand a separate extensive investigation.

We would like to thank again Referee #1 for his/her time and effort and helpful comments.

Sincerely,

Vasiliy Tikhonov and Co-author

**References**

Choudhury, B. J., Schmugge, T. J., Chang, A., Newton, R. W.: Effect of Surface Roughness on the Microwave Emission From Soils, J. Geophys. Res., 84, C9, 5699-5706, https://doi.org/10.1029/JC084iC09p05699, 1979.

Pulliainen, J., Grandell, J., Hallikainen, M.: HUT snow emission model and its applicability to snow water equivalent retrieval, IEEE Trans. Geosci. Remote Sens., 1999, 37, 3, 1378–1390.

Sharkov, E. A.: Passive Microwave Remote Sensing of the Earth: Physical Foundations, Springer/PRAXIS, Berlin, Heidelberg, London, New York etc, 2003.

Tikhonov, V. V., Boyarskii, D. A., Kitaev, L. M., Raev, M. D., Cherenkova E. A.: Regional Features of Microwave Radiation and Snow Cover Interaction on the Example of the North of the European Part of Russia, in: 10th Specialist Meeting on Microwave Radiometry and Remote Sensing of Environment, Firenze, Italy, 11-14 March 2008, report 152, https://doi.org/10.1109/MICRAD.2008.4579512, 2008.

Tikhonov, V. V., Boyarskii, D. A., Repina, I. A., Raev, M. D., Sharkov, E. A., and Alexeeva, T. A.: Snow Cover Effect on Brightness Temperature of Arctic Ice Fields Based on SSM/I Data, in: Pr. Electromagn. Res. S., Stockholm, Sweden, 12-15 Aug. 2013, 514–518, 2013.

Tikhonov, V. V., Boyarskii, D. A., Sharkov, E. A., Raev, M. D., Repina, I. A., Ivanov, V. V., Alexeeva, T. A., Komarova, N. Yu.: Microwave Model of Radiation from the Multilayer "Ocean-atmosphere" System for Remote Sensing Studies of the Polar Regions, Prog. Electromagn. Res. B, 59, 123–133, https://doi.org/10.2528/PIERB14021706, 2014.

Tikhonov, V. V., Raev, M. D., Sharkov, E. A., Boyarskii, D. A., Komarova, N. Yu.: A model of microwave emission of Antarctic snow-firn layers, Sovremennye problemy distantsionnogo zondirovaniya Zemli iz kosmosa, 14, 1, 200-204, https://doi.org/10.21046/2070-7401-2017-14-1-200-204, 2017. (In Russian).

---

## Author Comment (AC2) · 3 May 2018

Review of paper tc-2018-27 Tikhonov et al. "Theoretical study of ice cover phenology at large freshwater lakes based on SMOS data"

We thank Referee #2 for carefully reading our manuscript and the comments. We attempted to provide answers to all his/her questions and make necessary corrections. The answers are below in red.

General
The study deals with investigating L-band passive microwave emissions over freshwater lakes from a theoretical perspective, employing a forward model to simulate brightness temperatures which are then compared to observations from the SMOS mission.

The subject is interesting as relatively few studies have focused on modeling the passive microwave signatures from lakes. There are none that I am aware of that focus specifically on L-band. Although at present the coarse spatial resolution of spaceborne passive microwave sensors at L-band ultimately constrains their applicability to a limited number of large lakes, the study is still relevant given the specific advantages of this frequency compared to higher bands. In this way, this study also has the potential of bringing new information to the ongoing discussion of an operational follow-on sensor to the SMOS and SMAP exploratory missions.

From a general viewpoint the study is well written, clear and easy to follow. However, insufficient detail is given on key factors regarding the model simulations; I am left wondering if some kind of tuning of the forward model has been done. If so this is fine but clear details should be given so other groups may seek to replicate the results. Furthermore, comparisons of model results to observational data are very qualitative, e.g. no numbers (bias, RMSE etc.) are given. These factors weaken the broader conclusions that can be drawn from this study.

See answers to the comments below.

An additional aspect that I feel the authors omit is employing the full potential of SMOS data by analyzing data from different incidence angles. This may have relevance especially concerning conclusions made about sensitivity to dry snow (see major comments below).

See answers to the comments below.

Last, the references should be expanded to acknowledge more modeling work done for lakes at other microwave bands and studies related to SMOS and the cryosphere. Authors now refer mainly to their own work (with the exception of papers by Kang et al., which are extensively cited). In light of the above the potential of the paper is not really achieved. I give several recommendations below to help make this happen; I would recommend the authors should carefully address these comments before publication.

See answers to the comments below.

Major comments
1. Introduction, line 23: Add appropriate reference to the SMOS mission, e.g. Kerr et al., 2010.

An appropriate reference (Kerr et al, 2010) is added.

2. Introduction, throughout: The authors should refer more broadly to recent publications on applying L-band for the cryosphere. Studies on L-band signatures from the Antarctica (Macelloni et al.,), sea ice (Kaleschke et al.) and snow cover (Schwank et al.) have been published in recent years.

As suggested, we've broadened the review of the literature on cryosphere studies in L-band in "Introduction". Among them are investigations of brightness temperature of Antarctica ice sheet (Macelloni et al., 2013, 2014); analysis of snow brightness temperature dependence on its wetness, density and ground permittivity (Schwank et al., 2015; Naderpour et al., 2017; Schwank and Naderpour, 2018a, 2018b); snow thickness retrieval over thick Arctic sea ice (Maaß et al., 2013); investigation of brightness temperature variations of Arctic sea ice (Heygster et al., 2009; Richter et

al., 2018); retrieval of Arctic sea ice thickness (Huntemann et al., 2014; Kaleschke et al., 2012, 2016; Richter et al., 2017; Tian-Kunze et al., 2014; Zhou et al., 2018).

3. Although not dealing with lakes as such, the study by Schwank et al. (2015) should be of particular relevance as they also present a model for snow cover at L-band which should at least be cited. I assume the model could also be easily applicable for lake ice. Similarly, there are some other works looking at lake ice signatures at AMSR frequencies beyond Kang et al., which could be added (see e.g. Kontu et al., 2014).

As suggested, we've added a short review of various models of radiative properties of snow, ice and other Earth covers. The most widely used are HUT (Helsinki University of Technology) (Pulliainen et al., 1999; Lemmetyinen et al., 2010) and MEMLS (Microwave Emission Model of Layered Snowpacks) (Matzler, Wiesmann, 1999; Wiesmann, Matzler, 1999). HUT was successfully used for analysis of brightness temperature of snow- and ice-covered lakes and wetlands derived from aerial and satellite microwave radiometry data (Gunn et al., 2011; Kontu et al., 2014). Modified versions of MEMLS were used for detailed analysis of L-band emission of freezing ground covered with snow as well as wet snow cover (Schwank et al., 2014; Schwank et al., 2015; Schwank and Naderpour, 2018a, 2018b).

Also, we've added references (Du et al., 2017; Gunn et al., 2011; Kontu et al., 2014) to works on lake ice signatures at AMSR frequencies.

4. Section 2.1: The authors use only an incidence angle of 42.5 degrees from the L1C data (actually a collection of snapshots from 40 to 45 degrees). Why was this angle chosen over others (available from nadir to 70 degrees)?

The seasonal dependencies of brightness temperature at large freezing lakes discussed in our manuscript were in fact revealed incidentally as a by-product of another research. It had no connection with freshwater lakes and required no less than twice a day measuring of upwelling microwave radiation at a fixed viewing angle over a widest possible area. Indeed, the SMOS data contains measurements at various viewing angles. However, the imaging pattern is such that the maximum number of measurements falls into the 40-45 degrees range (Kerr et al, 2010). At the other angles, the resulting data time series have significant amount of gaps. From this perspective, the algorithms of streamline processing of L1C products were developed

and optimized so as to obtain as complete as possible time series of the data. Investigation of seasonal variations of brightness temperature at freezing lakes is undoubtedly of great interest. However, it requires certain amount of calculations and will be performed as a separate task.

5. Related to the above: Incidentally, SMOS incidence angles close to 41 degrees were noted by Schwank et al. (2015) to be insensitive to dry snow (whereas observations at other incidence angles were found to be sensitive to the presence of dry snow). Close to 41 degrees sensitivity to dry snow cover disappears at V-polarization, due to the opposing effects of impedance matching and refraction. Perhaps this is the cause the authors see no change in observed or modeled Tb with dry snow cover, as stated on p12, section 4? This should be discussed. Looking at other incidence angles beyond 42.5 degrees might shed some light here.

In our work, we attempt to explain seasonal brightness temperature variations observed at a viewing angle of 42.5 degrees at large freezing freshwater lakes. The reason for choosing this angle is given above. Referee #2 is quite right: recently, there have appeared a considerable number of works dedicated to Earth cover investigation in L-band. They regard Antarctica (Macelloni et al., 2013, 2014), snow cover (Maaß et al., 2013; Naderpour et al., 2017; Schwank et al., 2015; Schwank and Naderpour, 2018a, 2018b), and sea ice (Huntemann et al., 2014; Kaleschke et al., 2012, 2016; Richter et al., 2017; Tian-Kunze et al., 2014; Zhou et al., 2018; Richter et al., 2018). We have revealed the phenomenon of dramatic and fast change in L-band brightness temperature at freshwater lakes that was not discussed in literature before. The aim of this work is to provide an explanation of the phenomenon, which is, in our view, successfully achieved. Investigation of brightness temperature angular dependencies for snow on freshwater ice is a separate big task that could be addressed in future.

6. Section 3 is quite long, I suggest to divide into subsections. Section 3.1 could present the model and relevant equations, while section 3.2 could present the simulation setup (use of data etc) specific for this study (i.e. starting from p8, line 18).

As suggested, we've divided Section 3 into two subsections. The model is described in Section 3.1, its modification in Section 3.2.

7. Related to the above: p10 lines 10-17: This does not really deal with results, but should be moved from section 4 to section 3.

This part is moved to Section 3.2.

8. Section 3, section 4 and Table 3: Judging from figure 5 an almost perfect match of model vs. observations is achieved but it is not perfectly clear if e.g., tuning of model input parameters (given in Table 1) was required to achieve this. For TR1 no tuning is possible from what I see, but what about volumetric wetness for ice in TR2 and snow and ice in TR3? A range is given in Table 3, but it is not clear if these were defined individually for each lake? If no tuning was required, the authors could also highlight this.

We've added the following explanatory paragraph at the end of Section 3.2:

"In the modeling, we used mean seasonal values of snow density, ice grain diameter, volumetric wetness of snow, ice porosity and size of air bubbles in ice. These characteristics were taken from various sources (Kuz'min, 1957; Gray and Male, 1986; Kotlyakov, 2000; Kozlov, 2000; Cuffey and Paterson, 2010; Barry and Gan, 2011; Singh et al., 2011). It was assumed that in the transitional period (from TR2 to TR3), a gradual increase in lake ice porosity, as well as snow and ice wetness took place. These parameters are listed in Table 3. They were almost the same for all the lakes."

9. P10 lines 5-7: Dry snow and ice are transparent at L-band yes, but they still affect the emitted brightness temperature from the surface beneath though impedance matching and a change in the refraction angle. See Schwank et al., 2015. I imagine these results are also applicable to the water-ice-snow system.

We quite agree with Referee #2 in that ice and snow cover influence lake brightness temperature, that is why we put "almost completely transparent" not "completely transparent". This issue is highlighted in Section 4 when discussing TR2. At the chosen viewing angle (see answers to Comments 4 and 5 for the choice reasoning), snow is transparent, while ice adds to the radiation emitted by the water surface.

10. P12, line 19-20: Again, the insensitivity to snow may be due to the choice of incidence angle (although at H-pol, something could perhaps be seen). I suggest the authors analyze the response vs. incidence angle e.g. for one lake comparing simulations with the TR1, TR2 and TR3 setting against SMOS observations (e.g., the average Tb for those periods at different incidence angles).

As already mentioned in the answers to Comments 4 and 5, this work is intended to explain the revealed phenomenon of dramatic and fast seasonal change in brightness temperature of large freshwater lakes. Investigation of brightness temperature angular dependencies for snow on freshwater ice is a separate big task that could be addressed in future.

11. Section 5: The conclusions section is too short. The authors should provide a more complete assessment of their study, including the main results.

As suggested, we've expanded "Conclusions" to provide a more complete assessment of our study and detailed description of the results.

Minor/Editorial

1. P8, lines 8 and 19, maybe elsewhere: I think the convention is to write "wavelength" as one word (not wave length). Furthermore on line 19, wavelength is not a synonym to frequency, as it is now implied by the parenthesis. Please rephrase.

Corrected.

2. P9, line 17: "Note that for both media: : :" Sentence seems incomplete.

Corrected.

3. P12, line 13: "most wet". Unclear what is meant. Almost wet? Mostly wet?

Corrected.

4. P12 lime 14: Typo: capitalized 'M' in 'cm'

Corrected.

5. P13, line 7: "even in the cold season"

Corrected.

We would like to thank again Referee #2 for his/her time and effort, comments and suggestions that helped improve our manuscript.

Sincerely,
Vasiliy Tikhonov and Co-author

[revised manuscript text omitted]

---

## Author Response (AR2)

Dear Dr. Lars Kaleschke!

Thank you so much for your interest to our manuscript, remarks and suggestions to improve it. We tried our best to answer all your questions and make necessary corrections. Below are our answers (in blue).

To me it is not clear what the ultimate aim of the study is. To learn more about ice phenology of the large lakes or to show that the microwave model agrees with observations? Or the retrieval (?) of phenological parameters?

The goal of or study was to explain the observed seasonal variation of  L-band brightness temperature at freshwater lakes. In addition, we wanted to show the advantage of low frequency (L-band) measurements against SSM/I, SSMIS, AMSR-E and AMSR2 in the investigation of the cryosphere of the Earth. This advantage lies in the penetration depth of the formation of radiation, which makes it possible to evaluate the characteristics of the medium and the processes occurring in it from a thicker layer. With this in view, we used our model called "Thermal emission of multi-layered non-isothermal media". The model had been previously successfully employed in calculations of  brightness temperature of sea ice, Antarctic snow/firn layers and snow cover. For the purpose of this work, the model was modified for freshwater lakes. We tried to explain our goals in Introduction and briefly in Conclusion.

It was not made clear what the advantages of the microwave model and the SMOS data as compared to AMSR-based methods.

The employed frequency range (L-band) allowed highlighting the state of ice and snow at greater depths compared to frequency ranges of AMSR-E and AMSR2. The theoretical analysis showed that L-band thermal emission was received practically from the whole lake ice column during both cold winter and melt periods. This makes it possible to remotely sense processes in ice and snow cover even in the melt period. For the first time SMOS MIRAS data were used to assess the phenology phases of freshwater lakes. Please refer to Section 4. We also added the corresponding explanation to Introduction and Conclusion.

 The overall presentation quality is only fair. Some details are missing, e.g. the weblinks are not enough for the reproducibility. The data and the model could be described in more detail. Equation 1 comes with little motivation and introduction. Data sources and processing steps (smoothing, time averages?) for the figures are not clearly described.

We added more details about the used web resources. Also, Section 2.2 contains references to literature with climatic and glaciology data on the study lakes. In this section, Figs. 2-4 present multi-year data averages for all lakes.

We tried to explain in detail the essence of our model with a reference to the publication with complete formula derivation procedure (see Sections 3.1, 3.2).

In the calculations of brightness temperature, model input parameters were meteorological and glaciological data of the study regions and lakes. They were collected from a number of sources (publications and websites of scientific organizations).  The collected multi-year data were averaged and approximated by smooth dependencies on the date of observation. See details in Section 2.2.

Brightness temperature was calculated for every day during the whole study period (2012-2013). So, no time averaging was performed. This is pointed out in Section 3.2.

The SMOS L1C grid cell size is about 15 km. Ice thickness cannot be uniform across such an area. So, in the calculations, brightness temperature was averaged over ice depth within ± 10 cm (Section 3.2).

We thank again the Editor for useful comments and improvement suggestions.

Important notice.  In the period from 25 July to 15 August, 2018, all authors of the manuscript will take part in a scientific expedition in the north of Russia, out of reach by email, telephone, etc. In case of any necessity regarding the manuscript, please inform us either before 25 July or after 15 August.

Thanks for your understanding.

Sincerely,
Vasiliy Tikhonov and Co-authors

**Anonymous Referee #1**
Submitted on 15 May 2018

We thank Referee #1 for the interest to the manuscript and the suggestion. We tried to make a corresponding revision in the manuscript. Our comment is below (in blue).

The authors have made necessary revisions and improved the manuscript significantly. There is one suggestion though. As my previous comment, I would recommend the authors discuss about the limitations of using climatology data for the model inputs and the assumptions made for effective temperature, lake ice thickness, standard deviations of ice grain and air bubble sizes, and surface roughness of snow. It is difficult to carry out measurements on all ice/snow parameters required for model inputs at large scales and over the satellite observation time, but the authors should make it clear to the readers that such in-situ data are necessary for more rigorous model validations.

In line with the suggestion, we added a discussion of the problem in the end of Section 4.

Many thanks once again for the attentive consideration of the manuscript.

Sincerely,
Vasiliy Tikhonov and Co-authors

---

## Author Response (AR3)

We thank the Editor for his interest to our manuscript and useful suggestions to improve it.

The question about the correlation length is not quite clear to us. Perhaps, the Editor meant the effect that a deviation in snow and ice parameters can have on the stability of model results. As we mentioned before, one of the goals of the work was to provide an explanation of abrupt seasonal changes in brightness temperature of the lake surfaces. To this end, for the model input parameters we used regional averages of snow and ice parameters. The study of model stability against variation of input parameters is a rather demanding task. We plan to address it in the near future.

In line with the recommendations of the Editor, we changed the data source links to lower level domain ones.

Also, Author contribution and Competing interests sections were added.

We thank once again the Editor for his useful comments.

Sincerely,
Vasiliy Tikhonov and Co-authors